# Equity in Cancer and Chronic Disease Prevention through a Multi-Pronged Network Intervention: Works-in-Progress

**DOI:** 10.3390/ijerph21020213

**Published:** 2024-02-12

**Authors:** Yamilé Molina, Edward Tsai, Yalemzewod Enqubahry, Eunhye Lee, Faria Siddiqi, Anna Gottesman, Emma Boylan, Kate Paz, Margaret E. Wright, Ekas Abrol, Saria Lofton, Sage J. Kim, Ajanta Patel

**Affiliations:** 1Division of Community Health Sciences, School of Public Health, University of Illinois Cancer Center, University of Illinois Chicago, Chicago, IL 60612, USA; yenqub2@uic.edu (Y.E.); elee247@uic.edu (E.L.); fsiddi28@uic.edu (F.S.); 2University of Illinois Cancer Center, University of Illinois Chicago, Chicago, IL 60612, USA; etsai3@uic.edu (E.T.); mewright@uic.edu (M.E.W.); eabrol2@uic.edu (E.A.); 3School of Public Health, George Washington Milkin Institute, Washington, DC 20037, USA; anna.gottesman@gwmail.com; 4Chicago Department of Public Health, Chicago, IL 60612, USA; emma.boylan@cityofchicago.org (E.B.); kate.paz@cityofchicago.org (K.P.); ajanta.patel@cityofchicago.org (A.P.); 5Population Health Nursing Science, College of Nursing, University of Illinois Chicago, Chicago, IL 60612, USA; slofto4@uic.edu; 6Division of Health Policy and Administration, School of Public Health, University of Illinois Cancer Center, University of Illinois Chicago, Chicago, IL 60612, USA; skim49@uic.edu

**Keywords:** chronic disease prevention, cancer risk reduction, social determinants of health, social capital, social network methods

## Abstract

The increasing rates of cancer incidence are disproportionately borne by populations that are ineligible for screening and historically marginalized populations. To address this need, our community-centered model seeks to catalyze the widespread diffusion of evidence-based information and resources (e.g., community-based organizations, federally qualified health centers) to reduce the risks of cancer, chronic disease, and other conditions. In this study, we tested whether improving personal health literacy (i.e., confidence in seeking information) and enabling successful information transfer (i.e., intention to share the specific information learned through the program) among community residents could contribute to greater diffusion intention (i.e., number of network members with whom residents plan to share information and resources). The current study used post-intervention surveys, which were administered to Chicago residents who were 18 years or older and had participated in the program. Among the 1499 diverse Chicago residents, improved personal health literacy was associated with greater diffusion intention (ORs = 2.00–2.68, 95% CI [1.27–4.39], *p* ≤ 0.003). Successful information transfer was associated with greater diffusion, especially for cancer and other chronic disease risk reductions (ORs = 3.43–3.73, 95% CI [1.95–6.68], *p* < 0.001). The findings highlight the potential gains for health equity through sustainable, scalable, multi-sectoral partnerships.

## 1. Introduction

Equity in cancer prevention is a critical public health priority. In general, common cancers (e.g., colorectal) and cancers with increasing incidence rates (e.g., thyroid) have established, targetable risk factors, including obesity, physical activity, tobacco smoking, and human papillomavirus infection [1,2,3,4,5,6,7]. Prevention is particularly important given the growing incident rates among younger populations, for which there are limited evidence-based screening guidelines [8,9,10]. Colorectal cancer (CRC) represents an excellent example. From 2010 to 2030, CRC rates are projected to increase by 140% among 29–49-year-olds, who are largely ineligible for CRC screening [8,11,12,13,14]. Minoritized populations (e.g., African Americans, Latines) suffer disproportionately from these alarming trends in cancer incidence, partially due to disparities in cancer risk factors (e.g., obesity, unhealthy diets, metabolic risks) and contributing social determinants of health (SDOH, e.g., persistent poverty, food insecurity) [15,16,17,18]. Large-scale initiatives that focus on equity in cancer prevention are, thus, urgently needed. Such initiatives have high potential for holistic improvements in population health, given the many known risk factors (e.g., physical activity) that can simultaneously reduce risk for cancer and other chronic conditions (e.g., diabetes, cardiovascular disease) [19,20].

A growing number of multilevel frameworks and evidence-based strategies exist to tackle the disparities in common, shared risk factors for cancer and chronic disease [21,22,23]. Figure 1 highlights the stakeholders, strategies, and intervention targets that are of interest for the current study.

Within underserved communities, there are key organizations that can successfully deliver cancer and chronic disease prevention. These organizations include community-based organizations (CBOs) and federally qualified health centers (FQHCs) [24,25,26,27,28,29] that are already established and embedded within communities. Scholars have recently begun to focus on strengthening the relationships between health and social services, in line with social capital frameworks [30,31]. Hyperlocal, dense ecosystems could directly contribute to bidirectional, sustainable referrals to resources at CBOs, FQHCs, and other organizations [32,33]. Such referrals could enable a greater number of entry points for community members and could allow for seamless coordination across multiple, often co-occurring, unmet health and SDOH needs.

For these organizations, bicultural, bilingual community health workers (CHWs) and patient navigators represent commonly used, robust, effective frontline staff for cancer prevention and control programs [28,34,35,36,37]. CHWs and navigators are trained to offer community members personalized cancer education and resource navigation. Recent programs have further worked to enhance organizational health literacy [38,39,40], defined by the CDC as “the degree to which organizations equitably ensure individuals to find, understand, and use information and services to inform health-related decisions and actions for themselves and others” [41]. Such work recognizes that community members are not passive recipients of education and navigation. Rather, community members likely seek information for themselves and individuals in their social networks, after initial encounters with CHWs and navigators [42,43]. Thus, building on train-the-trainer models, CHWs and navigators can subsequently equip and train community members to become change agents to family, friends, and others [44,45].

Toward this goal, there are two potential mechanisms that may enable the successful, widespread diffusion of evidence-based information and resources. First, CHWs and navigators can spark information diffusion through enhancing personal health literacy skills among residents. Here, we use the CDC definition for personal health literacy [41], which is “the degree to which individuals have the ability to find, understand, and use information and services to inform health-related decisions and actions for themselves and others”. Building these skills is critical to ensuring that community members can seek and access health evidence and resources, beyond a single encounter with a CHW or navigator. Enhancing residents’ skills to seek knowledge may further allow for health-protective spill-over effects, including their ability to seek information that they can share with and for their family and friends [42]. A second, related mechanism for widespread network diffusion is information transfer, which we define here as the receipt of and intention to share evidence-based information (e.g., the role of smoking for lung cancer, cancer screening guidelines) [42]. Successful information transfer reflects both the quality of learning by the community member and the specificity of the message that they will share as a change agent. High information transfer thus may lead residents to use information for themselves and to diffuse high-quality messages throughout their networks [46,47,48,49]. While promising, these conceptual mechanisms may depend on various factors. For example, the decision to share health information widely, especially as a community member without formal training, may depend on the cultural perceptions of specific health topics (e.g., stigma about COVID-19 [50,51]).

Altogether, there are multiple, interlocked layers within communities that offer rich assets for the achievement of equitable risk reductions in cancer and other chronic diseases. The current study seeks to describe a community-centered model guided by the framework described above. Our analysis is guided by two objectives. First, we test our framework by examining the associations between improved personal health literacy skills and successful information transfer with increased intention to share messages within one’s network (network diffusion intention) among community members receiving this multi-component intervention. Our primary hypothesis is that improved personal health literacy skills and successful information transfer will be positively associated with the intention to share messages with a greater number of individuals within the network. Second, we examine the effectiveness of this approach for messaging about cancer and chronic disease risk reduction relative to other public health priorities (e.g., infectious disease/vaccines, emergency care). A secondary hypothesis is that the associations of personal health literacy skills and successful information transfer may be greater for cancer and chronic disease risk reduction than for other public health topics.

## 2. Materials and Methods

Context and Setting: Beginning in November 2022, a multilevel model was initiated by the Division of Chronic Disease Prevention and Health Promotion at the Chicago Department of Public Health (CDPH) to establish a community outreach and resource initiative to improve health equity in underserved communities, entitled the Community Health Response Corps (CHRC) Program. A key feature of the CHRC program is to connect residents to hyperlocal social and health resources and information (e.g., food pantries, preventing cancer through healthy diets), using a workforce of CHWs. The CHRC has been implemented across 17 Chicago Community Areas (Figure 2). Communities are situated in Chicago’s South and West Sides, wherein residents are predominantly Black/African Americans and Latine/Hispanic [52]. These communities were prioritized in 2022 by the City of Chicago, based on social–economic and health epidemiologic indicators, as part of the City’s strategic focus and investment in improving health and racial equity [53].

**Program:** The current study focused on one programmatic focus for the CHRC, which concerns improving health literacy. The project was funded by the Office of Minority Health within the U.S. Department of Health and Human Services (Award# CPIMP211238-01). The University of Illinois at Chicago’s (UIC) Institutional Review Board approved this study as exempt (STUDY2022-1367; 7 Nov 2022). The current study focused on interim findings from a larger program evaluation, focusing on program activities that occurred between March and November 2023. These evaluation activities and the current study relied on cross-sectional, self-report data, including residents’ perceptions that their health literacy skills had improved, through the program, and their intention to disseminate skills, information, and resources to their networks.

**The Community Health Response Corps (CHRC) Program:** The mission of the CHRC program was to connect residents on the South and West Sides of Chicago to social and health resources (e.g., 211 hotline; navigation of quality, affordable, accessible care), with the goal of improving residents’ overall health. The current study leveraged one initiative that the CHRC Program is leading, in the context of health literacy.

Stakeholders. The CHRC Program deployed an equity-driven “hub-and-spoke model” [54]. The Chicago Department of Public Health (CDPH) was the funder and the lead for the strategic oversight of the program. There were two FQHC/safety net-affiliated research centers, which trained CHWs, facilitated meetings across organizations, and served as evaluators. There was a central organization that represented and facilitated program administration across 11 CBOs. Across the CBOs, there were 120 CHWs who served as hyperlocal interventionists for residents within each community area.

Multilevel Components (Figure 1). The CHRC components equipped CBO leaders, CHWs, and community residents with evidence-based information and the skills to diffuse within their networks and ensure that residents obtain access to care.

Coalition Building Activities With Organizational Leaders. Eleven full-time, established supervisors for CHWs participated in monthly virtual meetings, which focused on updates on CHRC program activities (e.g., upcoming training for CHWs) and relevant data (e.g., visualization of evaluation—Figure 2; most recent data on local health disparities); CBO information exchange about their mission, services, and referral processes; and shared decision making and brainstorming program activities (e.g., streamlined evaluation processes, upcoming health fairs). Intervention targets for this component were the establishment of meaningful linkages and genuine relationships between CBOs, CDPH, and FQHC/safety net systems.

Train-the-Trainer Activities With CHWs. The 120 CHWs across the 11 CBOs participated in separate monthly virtual meetings, which focused on exposure to evidence-based health information (e.g., diets to reduce risk of cancer, heart disease, diabetes), skills training, case studies, and requests for technical assistance. The primary intervention target for this component was organizational literacy, with a focus on the competencies and practices of CHWs. Here, the specific intervention target was the degree to which CHWs are equitably trained to equip community residents with the skills to find, understand, and use health information and resources for themselves and their networks. Ultimately, the goal of the train-the-trainer activities for the CHRC was to ensure that CHWs had the knowledge of resources and the ability to train residents as change agents.

Empowerment Activities With Residents. Community residents encountered program-based CHWs and CBOs through community health fairs, town halls, and other programming. During encounters, CHWs highlighted resources to enhance health literacy skills (e.g., how to use 211 hotline to access health resources); deployed plain language, teach-back methods, and other strategies to ensure the high quality of learning evidence-based education across levels of literacy and health literacy; and worked with community members to identify “take home” actions for themselves as recipients and as change agents that will support their family and friends [55,56]. The type of public health message (e.g., cancer, heart attack) shared by the CHW was largely based on residents’ self-identified needs. Messages also aligned with the central themes of the health events (e.g., insurance enrollment) wherein the encounter occurred, as was applicable. Intervention targets for this component included improved personal health literacy skills and the successful transfer of information to community residents.

**Design:** The larger program will leverage quasi-experimental and stepped wedge designs to compare intervention effects. For the current study, we focused on variations among communities with intervention activities.

**Data Sources:** The current study leveraged original data collection, which was augmented with publicly available datasets.

Original Data Collection.

Procedures. After the Empowerment Activities, the CHWs invited community members to participate and complete a one-time, 11-item, non-identifiable survey. Given the nature of this field study, response rates were not systematically tracked. For the current evaluation, the eligibility criteria for community residents included (1) interaction with a program-based CHW; (2) being 18 years or older; and (3) being a resident of Chicago. During recruitment, CHWs briefly shared standard consent language (e.g., voluntary nature of evaluation, privacy/confidentiality; benefits and risks of participation). Interested respondents could complete non-identifiable surveys in their preferred language (English, Spanish) and selected survey modality (online, paper). Paper surveys were immediately returned to the evaluation team for data entry and management, using REDCap electronic data capture tools hosted at the University of Illinois Chicago. Alternatively, participants could directly enter their data in REDCap, using a QR code provided by the CHWs. There was no compensation for participation in the evaluation.

Original Data Collection Variables.

**Chicago Community Area** was determined through respondents’ open-ended responses about their residence. Approximately 91% (1358/1499) of respondents self-reported their Chicago Community Area. The other 9% (141/1499) of respondents reported their zip codes instead; we cross-linked the zip code to the appropriate corresponding Chicago Community Area.

**Study Reach** was tracked at the community level. This measure included the number of community residents who received information from CHWs (Empowerment Activities of the program) and who completed surveys. Based on our preliminary review of the frequency distributions, we classified study reach as Community Areas with 1–34 survey respondents; 35–83 respondents; and 84 or more survey respondents.

**Language** was determined based on respondents’ preferences for the completion of surveys in English or Spanish.

**Data Collection Modality** was tracked by study staff as either a paper or online survey.

**Lessons Learned from CHWs and Multiple Lessons Learned from CHWs** focused on the evaluation of the Empowerment Activities and used an open-ended question (“During our conversation, the most important health topic we talked was…?”), previously piloted by the study team [42,43]. Multiple study team members were trained and coded responses via eight dummy codes, for each question, to reflect the major topics covered through this program—SDOH resources (e.g., food pantries), infectious disease/vaccines (e.g., COVID, flu), emergency care (e.g., heart attacks, overdoses), mental health (e.g., depression, anxiety), healthcare access/use (e.g., insurance enrollment, wellness visits), and cancer/chronic disease prevention. Cancer/chronic disease prevention topics included common cancer risks (e.g., diet, physical activity, obesity, tobacco cessation) and conditions/metabolic risk factors for cancer (e.g., blood pressure/hypertension, diabetes, COPD). For participants who self-reported more than one health topic, “Yes” was assigned for having Multiple Lessons Learned from CHW.

**Message to Share and Multiple Messages to Be Shared** focused on the evaluation of the Empowerment Activities and used an open-ended question (“What information or message will you share?”), previously piloted by the study team [42,43]. Multiple study team members coded responses via the same eight dummy codes used for Lessons Learned. For participants who self-reported their intention to discuss more than one health topic, “Yes” was assigned for having Multiple Messages to Share.

**Improved Personal Health Literacy Skills** was an intervention target for the Empowerment Activities and was a primary predictor for this study’s analyses. It was determined through a brief adapted version of the Calgary Charter on Health Literacy scale [57]. Sample items included “I feel more confident to find, get, and use health information” and “I feel more confident to communicate about health to others”. Respondents could respond either No (0) or Yes (1). Initial composite scores represented the sum of endorsed items. Based on the initial frequency distributions, we dichotomized the final composite scores for improved personal health literacy, such that respondents were assigned “No” if they endorsed less than the six items endorsed and “Yes” if they endorsed all six items.

**Information Transfer** was an intervention target for Empowerment Activities and was a primary predictor for this study’s analyses. It was measured by assessing the concordance in the Lessons Learned from CHW and Message to Share variables. Residents were asked what they had learned from the CHWs (Lessons Learned, e.g., “the relationship between cigarette smoke and lung cancer”). They were also asked what messages they intended to share with others (Message to Share, e.g., “Direct cigarette smoke and 2nd hand smoke can contribute to lung cancer and there are aids one can use to quit smoking”). If a participant’s responses were concordant between these two variables (e.g., responses to both items demonstrated knowledge of the connection to cigarette smoke and lung cancer), they were assigned “High” for information transfer. If their responses were not concordant (e.g., response to at least one item talking about lung cancer, but not cigarette smoke), they were assigned “Low” for information transfer.

**Network Diffusion Intention** was the primary short-term outcome for this study. It was determined using a single item that asked “How many people will you speak to about this information [Lessons Learned from CHW]?” [58]. Participants’ responses to this question were used as the measure of their intention to diffuse the information through their social networks.

Community Area Data Augmentation.

Procedures. The study datasets were augmented with the most recent publicly available data at the Chicago Community Area level. Chicago Community Area served as the unit of analysis and variable for data linkages. These community-level data were included as proxies for respondent-level covariates that were not collected, given the pilot nature of the study and efforts to minimize the uncompensated respondent burden.

Community Area Variables.

**Population Median Age** was calculated as the median age of all residents within each Chicago Community Area, averaging over a 5-year timeframe (2017–2021). Data were obtained from the American Community Survey. Based on preliminary frequency distributions, we classified the data into the following three tertiles: median age as 24.90–33.90 years old; 33.90–37.50 years old; and 37.50–48.70 years old.

**Population Size** was calculated as the count of all residents within each Chicago Community Area, averaged over a 5-year timeframe (2017–2021), using American Community Survey data. Based on preliminary frequency distributions, we classified the data into the following three tertiles: population size as 2279–26,482 residents; 26,482–39,381 residents; and 39,381–102, 608 residents.

**Area Deprivation Index (ADI)** was calculated at the census block level and aggregated to the Chicago Community Area. The ADI is an existing factor-based composite measure of socioeconomic deprivation, available in a national (1–100 scale) version, with higher values indicating higher deprivation [59,60]. The ADI is constructed from 17 variables in the domains of income, education, employment, and housing quality, collected by the American Community Survey and aggregated to U.S. Census block groups [59,60]. Based on preliminary frequency distributions, we classified the data into the following three tertiles: 9.77–59.89; 59.89–73.78; 73.78–92.23.

**Racial/Ethnic Diversity Index** was calculated for each Chicago Community Area, using 2017–2021 race- and ethnicity-specific population counts from the American Community Survey. This score is an established composite measure that reflects the probability that any two residents of an area, chosen at random, belong to different racial and ethnic backgrounds [61]. A score of 0 represents a perfectly homogeneous community (e.g., 100% African American), whereas a score of 1 indicates that almost everyone has different racial and ethnic characteristics. Based on preliminary frequency distributions, we classified the data into the following three tertiles: 0.00–0.18; 0.18–0.34; 0.34–0.67.

**FQHC/CBO Quantity** was used to estimate existing social capital within communities, at baseline, before the CHRC program was deployed. The variable was calculated as the number of FQHCs and CBOs within each Chicago Community Area, based on 2020 data from the Centers for Medicare and Medicaid Services as providing comprehensive primary and preventive care to medically underserved areas and populations, regardless of the ability to pay. Based on preliminary frequency distributions, we classified the data into the following three tertiles: 0–1 FQHC/CBOs; 2–5 FQHC/CBOs; and 6+ FQHC/CBOs.

**Analytic Plan**: Analyses were conducted using SPSS 29.1. We first described our sample, including the final analytic sample (Figure 3) and the spatial distribution, using the Chicago Health Equity Zones (Figure 2; [62]). Next, we reported descriptive statistics (Table 1). Given the low level of missingness, we used listwise deletion techniques. For our first objective, we examined the associations of the intervention targets (information transfer, improved personal health literacy skills) with our primary outcome (network diffusion intention). Given that the primary outcome was ordinal (0–1 network member, 2–3 network members, 4+ network members), we first examined the potential to conduct ordinal regression. Our analyses did not meet the assumption of proportionate odds (*p* = 0.001). Given this, we conducted a multinomial regression model with network diffusion intention as our outcome variable (referent category: 0–1 network member); information transfer and improved personal health literacy skills as our primary predictors; and various a priori conceptually selected covariates (data collection modality, preferred language, message type, study reach, population median age, population size, ADI, racial/ethnic diversity index, FQHC/CBO quantity). For our second objective, we examined the moderating effect of the lesson learned (cancer/chronic disease prevention message, other public health message) on the associations between the intervention targets (information transfer, improved personal health literacy skills) and our outcome (network diffusion intention). Based on standards for moderation analyses [63], we conducted a model that included the aforementioned covariates, primary predictors (information transfer, improved personal health literacy, message type), and two interaction effects (message type x information transfer, message type x improved personal health literacy). For significant interactions, we conducted subsequent stratified analyses by message type (cancer/chronic disease prevention message, other public health message). Support for moderation was determined by the significance of the interaction terms. Finally, we conducted two sets of sensitivity models. First, given the program’s focus on minoritized populations and residential segregation in Chicago, we replicated the models including only residents from the West and South Sides of Chicago. Second, we replicated the models using multiple imputation by chained equations with 20 iterations [64]. For all analyses, an alpha of 0.05 was used to determine statistical significance.

## 3. Results

### 3.1. Study Sample Characteristics

Across nine months, the CHWs interacted with 1725 residents (Figure 2 and Figure 3; Table 1), who participated in the intervention and completed surveys. In line with the CHRC mission, 92% of respondents lived in the South and West Sides of Chicago. Regarding eligibility, 71 residents self-reported living outside of Chicago and were excluded from the analyses. Overall, there were relatively low levels of missingness (12%). The variables with missingness were the self-reported community area (*n* = 84), improved health literacy skills (*n* = 66), and the health topic that they discussed with their CHWs (*n* = 49). For missingness, primary models were conducted with a final analytic sample of 1499 residents with complete data. Sensitivity models were conducted with multiple imputation techniques.

Table 1 depicts the study sample characteristics. Approximately half of the sample completed the evaluation surveys online (49%) and half completed paper surveys (51%). Most residents completed the surveys in English (81%). The sample had a greater proportion of Spanish speakers (20%) relative to the city at large (2018–2022: 7%) [65]. Approximately 28% of respondents lived in community areas wherein there were at least 84 or more residents who participated in the program and completed surveys. Regarding represented neighborhood characteristics, roughly one third of residents lived in communities wherein the median population age was 38 years or older; the population size was 39,381 residents or more; the area deprivation index was 73.78 or more; and there were six or more FQHCs/CBOs embedded in their community area.

Most encounters between residents and CHWs focused on a single message (83%). The most discussed information was SDOH resources (36%); cancer and chronic disease prevention (26%); and infectious disease and vaccines (25%). Regarding intervention targets, 91% of respondents self-reported improved personal health literacy skills and 71% demonstrated high information transfer (i.e., same information learned and to be shared), after the CHW encounter. About 89% of respondents indicated that they intended to share the information with their network members. Of these, 32% of respondents reported that they would share information with four or more network members.

### 3.2. Objective #1: Associations between Information Transfer, Improved Personal Health Literacy Skills, and Network Diffusion Intention

Table 2 depicts the findings from our primary multinomial regression model, which included several covariates (data collection modality, preferred language, message type, study reach, population median age, population size, ADI, racial/ethnic diversity index, and FQHC/CBO quantity). Respondents who self-reported greater improvements in health literacy after interacting with a CHW had greater odds of intending to engage with 2–3 network members (30% vs. 24%; OR = 2.00, 95%CI [1.27, 3.13], *p* = 0.003) and 4+ network members (33% vs. 18%; OR = 2.68, 95%CI [1.64, 4.30], *p* < 0.001) than 0–1 network members, relative to other respondents. Similarly, respondents with high information transfer had greater odds of intending to engage with 2–3 network members (32% vs. 25%; OR = 2.27, 95%CI [1.66, 3.10], *p* < 0.001) and 4+ network members (35% vs. 24%; OR = 1.84, 95%CI [1.37, 2.47], *p* < 0.001) than 0–1 network members, relative to other respondents.

Among the covariates, network diffusion intention was also associated with completing paper surveys <0.001); the Spanish language (ps ≤ 0.04); learning about cancer and chronic disease prevention (ps < 0.001); living in less disadvantaged communities (only 0–1 vs. 2–3 network members: *p* = 0.02); living in less racially/ethnically diverse communities (only 0–1 vs. 2–3 network members: *p* = 0.003); living in community areas with a younger population median age (*p* ≤ 0.02); and living in community areas with fewer FQHCs (only 0–1 vs. 4+ network members: *p* = 0.03).

### 3.3. Objective #2: The Moderating Role of Message Type

Next, we conducted a multinomial regression model to test moderation, including aforementioned covariates, primary predictors (information transfer, improved personal health literacy, message type), and two interaction terms (message type * information transfer, message type * health literacy). There was evidence to suggest a moderating role of the message type for information transfer (ps = 0.04–0.009), but not for improved personal health literacy skills (ps = 0.55–0.71).

Table 3 depicts abbreviated findings for the subsequent stratified analyses, which adjusted for covariates and improved personal health literacy. Information transfer was significant across different levels of network diffusion intention (ps < 0.001) for participants who learned cancer and chronic disease prevention information from CHWs. This relationship was only significant for participants who learned other public health information when comparing the intention to share with 0–1 vs. 4+ network members (*p* = 0.03). In addition, the magnitude of associations appeared stronger for participants who learned cancer and chronic disease prevention information, based on a qualitative comparison of the odds ratios.

### 3.4. Sensitivity Analyses

The sensitivity analyses replicated the multinomial regression models above, respectively, (1) focusing on participants from West and South Chicago only (*n* = 1371) and (2) including the entire sample, by using missing imputation with chained equations for 20 cycles (*n* = 1654).

Among West and South Chicago participants, network engagement intention was associated with improved health literacy skills (ps ≤ 0.004) and high information transfer (ps ≤ 0.002). As in the primary models, interactive effects were observed for information transfer (ps = 0.003–0.005), but not for health literacy (*p* = 0.31–0.89). The stratified analyses revealed stronger associations between information transfer and network diffusion intention among respondents who learned cancer and chronic disease prevention information (ps ≤ 0.001) than respondents who learned other information (ps = 0.06–21).

Our second set of sensitivity analyses relied on multiple imputation techniques to handle missingness. Pooled analyses revealed similar patterns as in the primary models, wherein improved health literacy was associated with greater network diffusion (ps ≤ 0.008). Information transfer was moderated by the message type, although this was attenuated (0 vs. 1–2 network members: *p* = 0.009; 0 vs. 3–4 network members: *p* = 0.11). In line with this, the stratified models revealed attenuated differences in the associations between information transfer and network diffusion engagement among respondents who learned about cancer and chronic disease prevention (ORs = 3.21–3.37, 95%CI [1.86–5.57]) and other respondents (ORs = 1.69–1.97, 95%CI [1.19–4.14]).

## 4. Discussion

Equitable risk reduction and prevention in cancer and other chronic diseases is a critical, challenging priority. The current study developed a unique community-centered model to achieve cancer and overall health equity in Chicago, guided by social capital [30,31] and ecological frameworks [21,22,23]. Toward this goal, the program sought to build capacity and leverage interlocking linkages across organizations, CHWs and navigators, and residents within historically marginalized communities. Below, we highlight the promising preliminary findings on the feasibility of our program and the value of our framework.

Our findings contribute to growing research on the feasibility of network-based models for equity in cancer prevention and control [42,43,48]. Training 120 CHWs resulted in the successful reach of 1499 community members within less than a year, in line with past research on CHWs and navigators [46,47]. Our findings further suggested that CHWs and navigators can empower residents to become change agents themselves. Most residents indicated that they would share the health information that they learned with at least one family member, friend, or other community member. Preliminary evidence implicated successful information transfer (71% of residents) and the potential for high-quality, resident-driven information diffusion.

Our conceptual framework (Figure 1) and preliminary findings further add value to the understanding of why and when such models may work. Our findings suggested that personal health literacy skills may have multilevel health effects, in line with more recent conceptualizations of health literacy at the organizational and community levels [38,39,40,66]. The regression models found associations between self-reported improvements in personal health literacy skills and a greater intention to share information with more individuals. Our findings thus suggest that equipping residents with the skills to seek information for themselves may have positive spillover effects for their family and friends. Residents with greater personal health literacy skills may be more likely to share evidence-based information and to use their new skills to support their loved ones in obtaining needed health information and resources.

Additionally, our findings implicated the value of information transfer as an intervention target for network-driven health promotion [42,48]. Specifically, information transfer was conceptualized as a proxy for how well information was learned and the quality of the message that would be shared throughout the network. We posit that this construct is critical in understanding the variation in diffusion and the ultimate effect of the intervention on network behavior change. Our regression findings suggested that information transfer—i.e., the quality of the information learned and the message to be shared—may be critical for the widespread diffusion of health information and resources.

Our moderation analyses further suggest that successful information transfer may be particularly useful for network diffusion in the context of cancer and chronic disease prevention. Multiple factors may have influenced these findings. First, messages to reduce the risk of cancer and chronic disease may be easier to disseminate for community residents than other health topics that are stigmatized by the community, including infectious diseases and mental health [50,51]. Second, strategies to reduce cancer and chronic disease risks may be perceived as information that may benefit more network members, given the prevalence of chronic diseases (e.g., cancer, heart disease, diabetes) and the ability to address risk factors shared across conditions (e.g., diet, exercise) [19,20]. Finally, the prevention and early detection of chronic disease has been the cornerstone of community health programs for decades [28,34,35,36,37]. Residents may be more likely to have observed and even benefited from other programs related to chronic disease prevention. Such experience may have better equipped residents to retain detailed information about chronic disease. Past experiences may also contribute to more positive perceived norms in promoting chronic disease prevention in communities. Altogether, these findings highlight the importance of considering cultural values, norms, and other factors when considering public health intervention strategies, especially those focused on network diffusion. More research is warranted to explore the different cultural and contextual factors that may contribute to the differential efficacy of community-centered network interventions across public health conditions.

Our study has several important limitations to consider. Our sample relied on convenience-based sampling, which limited the generalizability. Relatedly, our study relied on CHWs to invite participants and administer surveys; however, there was not a systematic tracking system in place to record response rates. Thus, our findings were limited to populations who participated in the program and completed surveys, which may have exhibited greater social desirability, satisfaction with the program, and other factors that could have contributed to our results. For several variables, open-ended data were used and coded by multiple study team members. However, the intercoder reliability was not formally assessed, which may have influenced our findings. This secondary analysis was further limited by the cross-sectional design, which restricted our capacity for causal inference. Only post-intervention self-report data collection was available, which was likely subject to social desirability bias and other biases. This study collected very limited individual-level data, including a lack of demographic data (e.g., age, income, race/ethnicity, etc.). This study focused on the intention for network diffusion as an outcome. However, intention is a necessary, but not sufficient, predictor of behavior. This study focused on a community-centered model that was intentionally designed to be holistic and tailored to individual community members’ needs. This flexible design, however, had challenges in its evaluation, given the information that the community members received (e.g., SDOH vs. emergency care) from CHWs may have differed due to unmeasured factors.

Limitations notwithstanding, this current study offers several future venues for research and practice. This proof-of-concept study highlights the value of future robust evaluations of this and other similar community-centered programs, including the use of quasi-experimental/experimental designs, longitudinal data collection, and multilevel data collection among participating organizations, CHWs and navigators, participants, and network members. For this program, future evaluation will include its impact on bidirectional linkages and referrals across organizations; organizational health literacy; and confirmed network diffusion of information, behavior change, and well-being. Relatedly, future research should replicate and expand the current study’s findings regarding potential mechanisms/mediators (e.g., quality of relationships between organizations) and moderators of program effects (e.g., neighborhood characteristics). Future research should explore, in addition, how this type of holistic program can be adapted to address specific cancer types that are common, increasing, and for which there are significant racial/ethnic disparities. Such programs may, for example, leverage existing clinical–community linkages, including FQHCs and CBOs within the same marginalized neighborhoods, to promote risk reduction and risk-appropriate cancer care.

## 5. Conclusions

The current study highlights an important community-centered model that can address a critical, challenging priority—equitable risk reduction and prevention in cancer and other chronic conditions. Toward this goal, we developed a testable, ecological framework for multi-pronged interventions that enable organizations, CHWs and navigators, and residents to become hyper-localized change agents within their own marginalized communities. Guided by this framework, this hub-and-spoke model comprises and leverages a local health department, CBOs, and FQHC/safety net systems. Our findings suggest that optimizing personal health literacy skills and quality information transfer may transform residents from recipients to active resources in marginalized communities.

## Figures and Tables

**Figure 1 ijerph-21-00213-f001:**
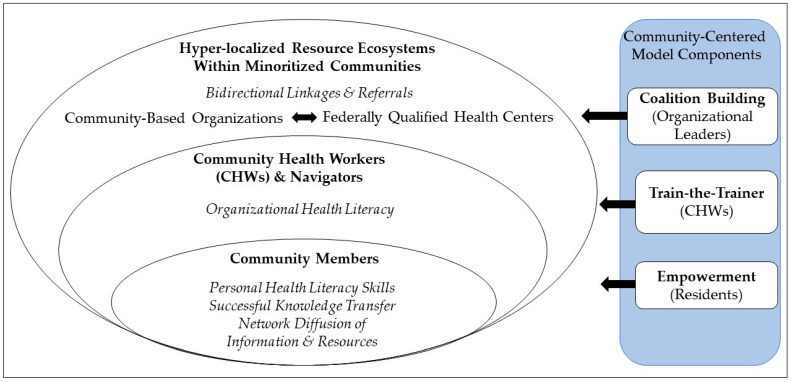
Conceptual model and community-centered model components.

**Figure 2 ijerph-21-00213-f002:**
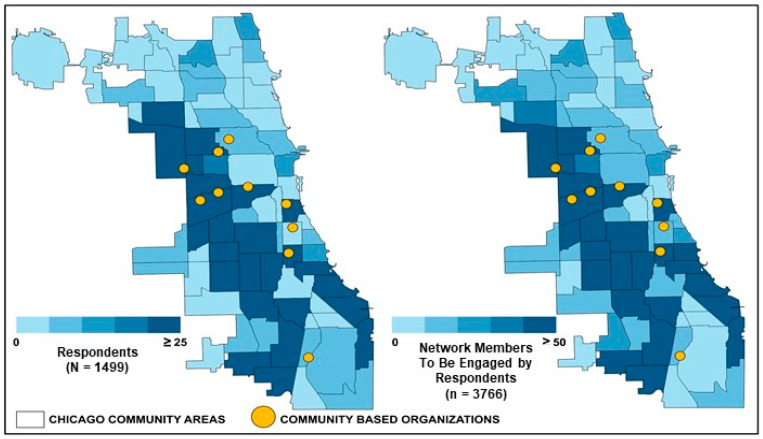
Spatial distribution of respondents and intended network diffusion.

**Figure 3 ijerph-21-00213-f003:**
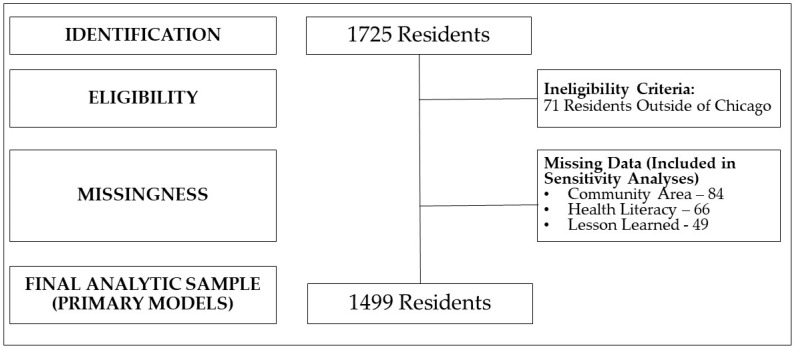
Study flow chart.

**Table 1 ijerph-21-00213-t001:** Study sample characteristics (*n* = 1499).

	*n*	%
*Respondent Characteristics*		
**Data Collection Modality**		
Online	758	49%
Paper	741	51%
**Language**		
English	1209	81%
Spanish	290	19%
**Lessons Learned from CHW**		
SDOH Resources (e.g., Food Pantry, Rent Support)	532	36%
Infectious Disease (e.g., COVID-19, Flu)	380	25%
Emergency Care (e.g., Heart Attack)	115	8%
Mental Health (e.g., Depression, Anxiety)	173	12%
Cancer Risks/Chronic Disease Prevention (e.g., Diet)	387	26%
Healthcare (e.g., Insurance, Primary Care Use)	103	7%
**Multiple Lessons Learned from CHW**		
No	1248	83%
Yes	251	17%
**Improved Personal Health Literacy Skills**		
No	137	9%
Yes	1362	91%
*Social Network Characteristics*		
**Messages to Share**		
SDOH Resources	517	35%
Infectious Disease	243	16%
Emergency Care	95	6%
Mental Health	123	8%
Chronic Disease Prevention	219	15%
Healthcare Access/Use	102	7%
**Multiple Messages to Share**		
No	1248	83%
Yes	251	17%
**Information Transfer**		
Low	441	29%
High	1058	71%
**Number of Network Members To Engage**		
0–1 network member	577	39%
2–3 network members	447	30%
4+ network members	475	32%
*Community Area Characteristics*		
**Geographic Region**		
North Side (Northwest, North Central)	128	9%
West Side (West, Southwest)	653	44%
South Side (Near South, Far South)	718	48%
**Study Reach within Community Area**		
1–34 survey respondents	508	34%
35–83 respondents	571	38%
84+ respondents	420	28%
**Population Median Age**		
≤33.90 years old	468	31%
33.90–37.50 years old	573	38%
≥37.50 years old	458	31%
**Population Size**		
≤26,482 residents	535	36%
26,482–39,381 residents	472	32%
≥39,381 residents	492	33%
**Area Deprivation Index**		
9.77–59.89	479	32%
59.89–73.78	590	30%
73.78–92.23	430	29%
**Racial/Ethnic Diversity Index**		
0.00–0.18	527	35%
0.18–0.34	494	33%
0.34–0.67	478	32%
**FQHC/CBO Quantity**		
0–1 FQHC/CBOs	505	34%
2–5 FQHC/CBOs	471	31%
6+ FQHC/CBOs	523	35%

**Table 2 ijerph-21-00213-t002:** Multinomial regression models to assess associations between intervention targets (health literacy skills, information transfer) and outcomes (network diffusion intention), adjusting for a priori covariates (*n* = 1499).

		Network Members					
Predictors	*n*	%	%	OR	95%CI		*p*-Value
		0–1 (REF)	2–3				
		(*n* = 577)	(*n* = 447)				
**Improved Personal Health Literacy Skills**				**2.00**	**1.27**	**3.13**	**0.003**
No (REF)	137	58%	24%				
Yes	1362	37%	30%				
**Information Transfer**				**1.68**	**1.26**	**2.25**	**<0.001**
Low (REF)	441	51%	25%				
High	1058	33%	32%				
Covariates							
**Data Type**				**2.00**	**1.52**	**2.62**	**<0.001**
Online (REF)	741	51%	27%				
Paper	758	26%	33%				
**Language**				**1.49**	**1.01**	**2.19**	**0.04**
English (Ref)	1209	42%	29%				
Spanish	290	23%	34%				
**Cancer Risks/Chronic Disease Prevention Information Learned**				**2.27**	**1.66**	**3.1**	**<0.001**
No (REF)	1062	44%	29%				
Yes	387	27%	37%				
**Study Reach within Community Area**				0.82	0.65	1.03	0.09
1–34 survey respondents	508	38%	32%				
35–83 respondents	571	32%	32%				
84+ respondents	420	48%	25%				
**Population Median Age**				**0.74**	**0.61**	**0.89**	**0.002**
24.90–33.90 years old	535	31%	33%				
33.90–37.50 years old	472	38%	31%				
37.50–48.70 years old	492	47%	26%				
**Population Size**				1.06	0.89	1.27	0.50
2279–26,482 residents	468	46%	28%				
26,482–39,381 residents	573	35%	27%				
39,381–102,608 residents	458	35%	34%				
**Area Deprivation Index**				**0.76**	**0.6**	**0.96**	**0.02**
9.77–59.89	479	38%	33%				
59.89–73.78	590	31%	31%				
73.78–92.23	430	49%	26%				
**Racial/Ethnic Diversity Index**				**0.72**	**0.59**	**0.89**	**0.003**
0.00–0.18	527	39%	31%				
0.18–0.34	494	40%	29%				
0.34–0.67	478	37%	29%				
**FQHC/CBO Quantity**				1.13	0.9	1.41	0.29
0–1 FQHC/CBOs	505	38%	27%				
2–5 FQHC/CBOs	471	38%	33%				
6+ FQHC/CBOs	523	40%	29%				
		**%** **0–1** **(*n* = 577)** **REF**	**%** **4+** **(*n* = 475)**	**OR**	**95%CI**		***p*-Value**
**Improved Personal Health Literacy Skills**				**2.68**	**1.64**	**4.39**	**<0.001**
No (REF)	137	58%	18%				
Yes	1362	37%	33%				
**Information Transfer**				**1.84**	**1.37**	**2.47**	**<0.001**
Low (REF)	441	51%	24%				
High	1058	33%	35%				
Covariates							
**Data Type**				**2.97**	**2.26**	**3.91**	**<0.001**
Online (REF)	741	51%	22%				
Paper	758	26%	41%				
**Language**				**2.14**	**1.46**	**3.12**	**<0.001**
English (Ref)	1209	42%	29%				
Spanish	290	23%	43%				
**Cancer Risks/Chronic Disease Prevention Information Learned**				**1.74**	**1.27**	**2.39**	**<0.001**
No (REF)	1062	44%	32%				
Yes	387	27%	36%				
**Study Reach within Community Area**				1.11	1.46	3.12	0.36 ^1^
1–34 survey respondents	508	38%	31%				
35–83 respondents	571	32%	36%				
84+ respondents	420	48%	27%				
**Population Median Age**				**0.72**	**0.60**	**0.87**	**<0.001** ^1^
≤33.90 years old	535	31%	36%				
33.90–37.50 years old	472	38%	32%				
≥37.50 years old	492	47%	27%				
**Population Size**				1.18	0.98	1.42	0.08 ^1^
≤26,482 residents	468	46%	25%				
26,482–39,381 residents	573	35%	38%				
≥39,381 residents	458	35%	31%				
**Area Deprivation Index**				1.00	0.79	1.27	0.99 ^1^
9.77–59.89	479	38%	30%				
59.89–73.78	590	31%	38%				
73.78–92.23	430	49%	25%				
**Racial/Ethnic Diversity Index**				0.98	0.79	1.21	0.85 ^1^
0.00–0.18	527	39%	29%				
0.18–0.34	494	40%	32%				
0.34–0.67	478	37%	34%				
**FQHC/CBO Quantity**				**0.78**	**0.63**	**0.98**	**0.03** ^1^
0–1 FQHC/CBOs	505	38%	35%				
2–5 FQHC/CBOs	471	38%	29%				
6+ FQHC/CBOs	523	40%	31%				

^1^ *p*-values testing for trend and descriptive statistics by tertile are presented to facilitate interpretability.

**Table 3 ijerph-21-00213-t003:** Stratified multinomial regression models to assess associations between intervention targets (information transfer) and outcomes (network diffusion intention), adjusting for improved personal health literacy skills and a priori covariates by message type (cancer and chronic disease prevention, other).

*Sub-Sample: Respondents Reporting Cancer and Chronic Disease Prevention Lessons Learned (n = 387)*				
Outcome: 0–1 network member (REF) vs. 2–3 network members				
Predictors	OR	95%CI		*p*-value
**Information transfer (REF: Low)**	**3.34**	**1.95**	**6.05**	**<0.001**
Outcome: 0–1 network member (REF) vs. 4+ network members				
**Information transfer (REF: Low)**	**3.73**	**2.08**	**6.68**	**<0.001**
*Sub-Sample: Respondents Reporting Other Public Health Lessons Learned (n = 1112)*				
Outcome: 0–1 network member (REF) vs. 2–3 network members				
Predictors	OR	95%CI		*p*-value
**Information transfer (REF: Low)**	1.31	0.93	1.84	0.12
Outcome: 0–1 vs. 4+ network members				
**Information transfer (REF: Low)**	**1.46**	**1.03**	**2.07**	**0.03**

## Data Availability

Interested parties should contact the corresponding author for data sharing of non-identifiable data.

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
