# Peer review of "Equity in Cancer and Chronic Disease Prevention through a Multi-Pronged Network Intervention: Works-in-Progress"

_ijerph, 2024, doi:10.3390/ijerph21020213_

Round 1

Reviewer 1 Report

Comments and Suggestions for Authors

---General comment

The study presents an excellent outline of its rationale, with great scientific language. Although the introduction is absent of a hypothesis, and does not explore the city where the study was conducted, it manages to lead the reader to understand the objective of the proposal. I have few considerations. In short, I believe that the graphic quality of images can be drastically improved.

 ----Specific comments

--Introduction

-Lines 65 and 75 – These two paragraphs begin with “Within”. Try to improve it.

-Line 94 – Remove quotes here.

-Line 106 – The terms “cultural contexts” are vague.

-Line 118 – After the objectives, provide the hypotheses.

--Methods

-Improve the graphical quality of Figure 2.

Author Response

The study presents an excellent outline of its rationale, with great scientific language. Although the introduction is absent of a hypothesis, and does not explore the city where the study was conducted, it manages to lead the reader to understand the objective of the proposal. I have few considerations. In short, I believe that the graphic quality of images can be drastically improved.

**We are thankful for the reviewer’s overall assessment and thoughtful considerations.

--Introduction

-Lines 65 and 75 – These two paragraphs begin with “Within”. Try to improve it.

**We thank the reviewer for this critique and have modified the wording, as recommended.

-Line 94 – Remove quotes here.

**We have removed the quotes, as recommended.

-Line 106 – The terms “cultural contexts” are vague.

**We have provided more detailed wording, as recommended, for this sentence.

-Line 118 – After the objectives, provide the hypotheses.

**We thank the reviewer for this important suggestion and now include hypotheses, as was recommended.

--Methods

-Improve the graphical quality of Figure 2.

**We now provide a high-quality image of Figure 2.

Reviewer 2 Report

Comments and Suggestions for Authors

Thank you for the opportunity to review this manuscript. I have some suggestions which I feel will improve the quality of the manuscript, as follows:

1. The abstract as a whole needs reorganisation in order to provide a clear background (e.g., why are interventions to improve equity needed?) and methods. Results and conclusions are fine as is, but it is unclear from this abstract what the actual study is about. As an example, line 29, why is this "Second"? What is first? This sentence as a whole is unclear. 

2. The introduction is well-written and provides a sound background to the study. Perhaps some of this information could find its way to the abstract to clarify the purpose of the study? 

3. Methods, page 5, line 181. It's unclear how improvements in health literacy can be measured in the current study given there is only a one-time survey collected. Is this an aim of the larger study rather than this publication? If so, this should be made clearer. If not, please explain how this will be achieved. I can see from later in the Methods that this is a self-report measure, but I think further information/clarification would be good. 

4. Methods, Study Reach. Could you clarify whether this refers to those who either participated in the activities or the data collection, or who participated in both? If the latter, could it be that the reach was greater than identified but people just did not answer the survey? What was the response rate? If the former, how was this data collected? 

5. Results, Figure 3 and line 332. Does this mean that all residents who interacted with the CHWs completed a survey? So the response rate was effectively 100%? Please clarify. 

6. Results, line 366-367. Is this number a simple addition of the survey responses? Could it not be that respondents were thinking of the same individual with whom they would share information, and so this number could in fact be (much) smaller? I would remove this sentence if so. 

7. Results, lines 378-382. I'm not sure of the importance of this finding, as the two questions are essentially gathering the same information in a different way and so one would expect a high degree of overlap between them. 

Minor comments:

1. Please ensure you explain abbreviations at first use - e.g. FQHC in Abstract, line 28. 

2. There are some minor spelling and grammatical errors throughout the manuscript.

Comments on the Quality of English Language

While this manuscript was generally well-written, there were some minor spelling and grammatical errors. 

Author Response

Thank you for the opportunity to review this manuscript. I have some suggestions which I feel will improve the quality of the manuscript, as follows:

  1. The abstract as a whole needs reorganisation in order to provide a clear background (e.g., why are interventions to improve equity needed?) and methods. Results and conclusions are fine as is, but it is unclear from this abstract what the actual study is about. As an example, line 29, why is this "Second"? What is first? This sentence as a whole is unclear. 

**We have re-organized the abstract, as thoughtfully recommended by the reviewer.

  1. The introduction is well-written and provides a sound background to the study. Perhaps some of this information could find its way to the abstract to clarify the purpose of the study? 

**We thank the reviewer for this excellent suggestion and have sought to incorporate the background information from the introduction into the abstract.

  1. Methods, page 5, line 181. It's unclear how improvements in health literacy can be measured in the current study given there is only a one-time survey collected. Is this an aim of the larger study rather than this publication? If so, this should be made clearer. If not, please explain how this will be achieved. I can see from later in the Methods that this is a self-report measure, but I think further information/clarification would be good. 

**We appreciate this suggestion. We have now included more detail about using cross-sectional and self-report data on perceptions of change / improvement in knowledge literacy in the referenced section.

  1. Methods, Study Reach. Could you clarify whether this refers to those who either participated in the activities or the data collection, or who participated in both? If the latter, could it be that the reach was greater than identified but people just did not answer the survey? What was the response rate? If the former, how was this data collected? 
  2. Results, Figure 3 and line 332. Does this mean that all residents who interacted with the CHWs completed a survey? So the response rate was effectively 100%? Please clarify. 

**We thank the reviewer for this important point. The current study trained CHWs to invite all participants to provide information and to complete surveys. To minimize CHW burden, we unfortunately did not train CHWs to track, systematically, the number of participants who did not agree to complete surveys. Given this, we do not have data on the response rate. We do agree that this is an important limitation for our pilot field study and have included detailed information about the limitation and caution that must be used when evaluating our findings, given this limitation.

  1. Results, line 366-367. Is this number a simple addition of the survey responses? Could it not be that respondents were thinking of the same individual with whom they would share information, and so this number could in fact be (much) smaller? I would remove this sentence if so. 

**We have removed this sentence, as recommended by the reviewer.

  1. Results, lines 378-382. I'm not sure of the importance of this finding, as the two questions are essentially gathering the same information in a different way and so one would expect a high degree of overlap between them. 

**We appreciate the reviewer’s concern. Our predictor variable, information transfer, is operationalized to measure the depth to which residents understood and could share specific public health messages. For example, participants with ‘high information transfer’ would state that they both learned and were interested in sharing information on the relationship between lung cancer and cigarette smoke. Conversely, participants with ‘low information transfer’ would report more general, non-concordant information when asked what they learned from the CHW (e.g., vaccines are important) and what they intended to share with network members (e.g., COVID is on the rise again).  Our predictor thus reflects the quality of learning from the CHW and the specificity of the message that the resident will disseminate. We have provided more clarifying detail to highlight the relevance of this construct in our Introduction, when describing intervention activities to improve information transfer, and when describing the measure and operationalization. Our outcome, diffusion intention, is operationalized to measure the number of network members with whom the resident hopes to engage. Thus, our study assesses if the quality of learning / specificity of the message retained by the community resident is associated with the intention to diffuse that message widely in the network. Our findings for this association are important, as they have implications for why certain train-the-trainer programs may not work, including reduced intention to disseminate among change agents, potentially due to limited learning, and less impact on network behavior, due to less specific information being diffused throughout community. We now include this information in more detail in the Discussion and are grateful for the reviewer’s suggestions and recommendations.

Minor comments:

  1. Please ensure you explain abbreviations at first use - e.g. FQHC in Abstract, line 28. 

**We have sought to explain abbreviations at first use, as recommended by the reviewer.

  1. There are some minor spelling and grammatical errors throughout the manuscript.

**We have sought to address minor spelling and grammatical errors, but would be happy to fix any errors we may have missed.

Reviewer 3 Report

Comments and Suggestions for Authors

Very valuable and easy to read publication. Another positive aspect is that the intervention was developed on the basis of a framework. After minor adjustments, I would recommend publication. 

Abstract: please use the full wording of FQHC at the first mention 

l. 146-151: think about rephrasing this part since it is hard to comprehend 

l. 169: I feel that the definition of organizational literacy falls a little short and it you target not all aspects of it. Please mention which aspects of organizational literacy you want to target with this intervention.

l. 407: typo in qualitative

Methods: Did you check the intercoder reliability? You mentioned that several stuy members coded the results. If not please mention this in the limitations. 

Discussion: The results of the regression models are not really discussed. Your insight into this would be really interesting. 

Author Response

Very valuable and easy to read publication. Another positive aspect is that the intervention was developed on the basis of a framework. After minor adjustments, I would recommend publication. 

**We are thankful for the reviewer’s overall assessment and thoughtful considerations.

Abstract: please use the full wording of FQHC at the first mention 

**Based on other reviewer comments, we have modified the abstract and no longer specify FQHCs in the abstract. Nonetheless, we have endeavored to ensure full wording of our abbreviations throughout the manuscript.

  1. 146-151: think about rephrasing this part since it is hard to comprehend 

**We thank the reviewer for this suggestion. We now have separate sentences to describe each stakeholder and their role within our program.

  1. 169: I feel that the definition of organizational literacy falls a little short and it you target not all aspects of it. Please mention which aspects of organizational literacy you want to target with this intervention.

**We appreciate this concern and have added more detail, highlighting our focus on organizational literacy in the context of the competency and practices of CHWs to deliver health information and train community residents to become change agents themselves.

  1. 407: typo in qualitative

**We have corrected the typo.

Methods: Did you check the intercoder reliability? You mentioned that several stuy members coded the results. If not please mention this in the limitations. 

**We now have added the lack of formal assessment of intercoder reliability as a limitation, as thoughtfully recommended by the reviewer.

Discussion: The results of the regression models are not really discussed. Your insight into this would be really interesting. 

**We thank the reviewer for this suggestion and have provided more discussion about our models and venues for research in the Discussion.